# The Significance of a Multilocus Analysis for Assessing the Biodiversity of the Romanov Sheep Breed in a Comparative Aspect

**DOI:** 10.3390/ani13081320

**Published:** 2023-04-12

**Authors:** Nurbiy S. Marzanov, Davud A. Devrishov, Mikhail Y. Ozerov, Oleg P. Maluchenko, Saida N. Marzanova, Elena B. Shukurova, Elena A. Koreckaya, Juha Kantanen, Daniel Petit

**Affiliations:** 1Center for Animal Husbandry Named after Academy Member L.K. Ernst, Podolsk-Dubrovitsy, 60, Moscow Region 142132, Russia; 2Moscow State Academy of Veterinary Medicine and Biotechnology Named after K.I. Skryabin, Federal State Budgetary Educational Institution of Higher Education, ul. Akademika Skryabina, 23, Moscow 109472, Russia; 3Biodiversity Unit, University of Turku, 20014 Turku, Finland; 4State Scientific Institution All-Russian Scientific Research Institute of Agricultural Biotechnology, UL Timiriazevskaya, 42, Moscow 127550, Russia; 5Federal State Budgetary Institution “Far Eastern Scientific Research Institute of Agriculture”, s. Vostochnoe, UL Clubnaya, 13, Khabarovsk 680521, Russia; 6Tver State Agricultural Academy, Federal State Budgetary Educational Institution of Higher Education, UL Marshala Vasilevskogo, 7, Sakharovo, Tver 170904, Russia; 7Natural Resources Institute Finland (Luke), 31600 Jokioinen, Finland; 8LABCiS, University of Limoges, UR 22722, F-87000 Limoges, France

**Keywords:** Romanov sheep, hemoglobin, transferrin, albumin, prealbumin, *BMP-15*, *BMPR1B*

## Abstract

**Simple Summary:**

The history of the Romanov breed goes back more than a century, with the first mention being made in 1802. This breed has unique qualities, such as a good resistance to harsh environments, high litter size, polyestrousity, early maturation and excellent fur coat qualities. In this work, we highlighted several genetic markers involved in these characteristics. The high ovulation rate of the Romanov breed is probably linked to the polymorphism recorded in two genes located on different chromosomes. The breed’s adaptation to the harsh environment of its breeding area in the Russian Federation could be associated to a high level of heterozygosity. In addition, the dilemma of its obscure origin was also addressed by comparison with other Russian breeds of different productive orientations.

**Abstract:**

The Romanov breed was evaluated using immunological and genetic markers. The seven blood group systems were characterized with a greater accuracy than in previous works on sheep in the Russian Federation, and were compared to eight ruminant species. Unlike other breeds, Romanov sheep shows a higher frequency of *HB^A^* than *HB^B^* alleles. There are 3–4 genotypes at the transferrin locus whereas in other breeds 6–11 genotypes have been found. At the albumin locus, the majority of the identified genotypes were heterozygotes, unlike in the other breeds studied. In the prealbumin locus, the Romanov breed was the only one where all the genotypes were heterozygous. We speculate that polymorphism at two loci (*BMP-15* and *BMPR1B*) could effect on the high ovulation rates of Romanov sheep. Based on different genetic markers, the prevalence of heterozygotes in the Romanov sheep could determine their higher viability. A cluster analysis showed the close proximity of 12 populations of the Romanov breed, as the breeding stock come from the Yaroslavl region.

## 1. Introduction

In the central zone and in the European north of the Russian Federation, the traditional Romanov sheep is one of the important breeds for meat production and fur coats. The Romanov breed is one of the 200 most famous sheep breeds in the world. Animals of this breed are characterized by exceptional fertility. Under normal conditions of feeding and rearing, 100 ewes give birth to 260–300 lambs per year, and the selection of the most prolific parental animals allows to breed this character [1,2]. The high litter size typically requires additional feeding for lambs as an ewe cannot feed the three to seven lambs born in the same litter. This sheep is considered a peasant sheep because its multiple pregnancies provide the family with work in the winter, related to the care of the lambs. Meat of the Romanov breed has typically a high quality. The skin has unique properties; it is used for the production of short fur coats, caftans and other leather products. The fleece of Romanov sheep is grey, due to a mixture of short black coarse fibers and white (or paler) finer fibers, with a ratio of secondary to primary follicles of 5.2 [3]. This fleece structure is very important for the long winters in Russia, because during the Great Patriotic War (1941–1945), it saved many soldiers at the frontline from the cold as they wore sheepskin coats. Therefore, the Romanov breed is often called the “war sheep”. In recent years, the breeding area of the Romanov breed, which originated in the central regions of the Russian Federation, has been extended to the Lower Volga region, Siberia, the Far East, the Northern Caucasus and even to several countries in Europe, North America, Iran and Turkey. Therefore, the preservation of this unique breed in Russia is of great importance.

Using various types of genetic markers (blood group systems, polymorphic proteins, microsatellites), a number of breeds bred in the territory of the former USSR were characterized [4,5,6,7,8]. We highlight the possibility of using these markers to assess the genotypes of several loci in order to (i) understand how these different types of genetic markers could be involved in the adaptation of the populations to the harsh environmental conditions of the Russian Federation, and (ii) contribute to an understanding of the origin of the breed, and (iii) help in decision-making regarding their production. To document the importance of the specificity of the Romanov breed, we carried out a comparative analysis with other sheep breeds and even other domestic ruminants from Eurasia.

## 2. Materials and Methods

Blood samples for the study of the Romanov breed were collected for detection of brucellosis and infectious epididymitis between 2015 and 2018, and have been used in previously published works [8,9]. No extra sampling was needed for the present study. The determination of blood groups of the 7 systems (A, B, C, M, D, R, I) was carried out according to standard methods [5]. For the 12 systems present in the ruminant studied, 66, 41 and 10 sera-reagents for bovine, caprine and ovine, respectively, were used. The 10 sheep serum reagents used in goats were anti-Aa, anti-Bb, anti-Bd, anti-Be, anti-Bi, anti-Bg, anti-Ca, anti-Ma, anti-Mb and anti-R. In the complex A, B, C, M systems of blood groups, the identifications of recessive and dominant alleles and calculations of genetic factors were deduced by the method of Bernstein [10], using Fisher’s maximum likelihood method. The data concerning the systems D, R and I were taken as presented in the work of Marzanov et al. [9]. The occurrence of alleles and genotypes at diallelic loci (D, R, I) was determined based on the Hardy–Weinberg equation [5]. To study the *BMP-15*, *BMPR1B* and *GDF9* genes linked to high ovulation rates, the blood samples were taken from 4–5 years old ewes. After DNA isolation from whole blood, the determination of normal and mutant alleles in the *BMP-15* locus was carried out according to the method of Polley et al. [11] and Malyuchenko et al. [12]. For this purpose, the sequencing of these genes was undertaken using an ABIPrism 3130xl device. Regarding the BMP-15 polymorphism concerning the mutations *FecX^G^* [391 (C-T)], *FecX^H^* 544 (C-T), *FecX^I^* [579 (T-A)], *FecX^L^* [635 (G-A)] and *FecX^B^* [773 (G-T)], primers defined in [13] were selected so that each amplified fragment contained the sites of all the single nucleotide substitutions described (SNPs, or single nucleotide polymorphism). In addition, the results for the *BMP-15* polymorphism in another population of Romanov sheep were taken from the work of Marzanov et al. [14].

The polymorphism of blood proteins in sheep from Russia, Azerbaijan and Moldova (hemoglobin, transferrin, albumin, prealbumin) was determined by an electrophoretic test in polyacrylamide gel and the subsequent staining of phoregrams [15]. All the results on the b hemoglobin locus in different sheep breeds are new, except for the population I of Romanov sheep published in Marzanov et al. [9]. Regarding the transferrin and prealbumin loci, the results are also new, except for the three populations of Romanov sheep previously published in Marzanov et al. [9]. The data on the albumin locus were entirely taken from the work of Marzanov et al. [9]. 

The method of Cavalli-Sforza and Edwards [16] implemented in Populations 1.2.32 software (http://bioinformatics.org/populations (accessed on 13.02.2011)) was used to establish the genetic relationships between 12 populations from 5 regions of the Russian Federation in the form of a dendrogram obtained from 7 blood group systems. The robustness of nodes was determined by 1000 bootstrap replicates. The statistical processing of some of the obtained data was conducted with Spearman, Student’s, Fisher’s exact test and *χ*^2^ tests [16,17,18].

To obtain a global view on the heterozygosity and allelic richness of the Romanov sheep compared to other Russian coarse wool breeds, we considered the data of 20 microsatellite markers published by Tapio et al. [8]. We calculated the residuals of these parameters against population sizes. The residuals of the group including the Romanov breed were, themselves, compared to the rest using a *t*-test. 

## 3. Results and Discussion

### 3.1. Characteristics of the Romanov Breed According to Seven Systems of Blood Groups

Table 1 presents a detailed description of the Romanov breed according to six blood group systems [9]. It shows data on phenotypic classes, genotypes and alleles. For each system, the number of individuals is in brackets.

The B blood group system is shown separately (Table 2).

According to the data obtained, the B blood group system is as complex in sheep as it is in cattle and goats. As can be seen in Table 2, it contains many more phenotypic classes, genotypes and alleles than the other A, C, M, D, R and I blood group system data above. System B distinguishes simple and complex alleles. Simples have six alleles (*B^b^*, *B^d^*, *B^e^*, *B^i^*, *B^g^*, *B^-^*) and complexes up to ten (*B^bde^*, *B^bdi^*, *B^bdg^*, *B^bei^*, *B^beg^*, *B^big^*, *B^dei^*, *B^deg^*, *B^dig^*, *B^eig^*). 

As this analysis was more comprehensive than previous studies done on Russian breeds, it is difficult to estimate the rate of difference between the allele frequencies found in the Romanov breed compared to other breeds. However, compared to the Ukrainian breeds Tsigai, Ascanian Merino and Karakul [19], it turns out that the pattern of allele frequencies in Romanov sheep is original, except in the case of D system, for which the *D*^-^ allele was always found more frequently than the *D^a^*.

The common antigen of blood groups for three species of ruminants (cattle/sheep/goat) was the erythrocyte factor named J for cattle and R for sheep and goat. In the diallelic J/R system in cattle, yak and goats, the relationship between *J^R^* and *J^r^* was dominant–recessive. In 15 heads of Romanov sheep among the 52 analyzed, it was impossible to determine the phenotypes R and O. As these animals were carriers of the genotype *I^i/i^* of the I system, this characteristic was interpreted as being due to epistasis [5,20]. At the same time, out of 66 bovine sera-reagents, 23 reacted with yak erythrocytes. Of the 41 goat serum reagents, only 3 were found to be common to sheep (anti-Ca, anti-R, anti-O). Only bovine and yak anti-J/R in triads reacted with sheep R antigen. At the same time, the 10 sheep serum reagents had positive results in goats. The serum-reagents of the blood groups of three ruminants (cattle, sheep and goats) did not react with the erythrocytes of camel, saiga, elk and muskox [21].

### 3.2. Diagnosis of High Ovulation Rate Genes in Sheep of the Romanov Breed

In the Romanov breed, the analysis was carried out to test for the presence of three genes involved in the proliferation and differentiation of ovarian follicles or granulosa cells: *BMP-15*, *BMPR1B* and *GDF9*. 

In the population of 50 Romanov sheep, the previous results of Marzanov et al. [14] were summarized; this is shown in Table 3. In the animals with the WW genotype, there was a higher percentage of non-productive and barren ewes than in individuals heterozygous for this locus; in individuals homozygous for the mutant allele, the percentage was minimal.

From these data, it can be seen that individuals with the MM genotype gave the highest number of lambs per year, both at the time of birth and weaning, with 3.0 ± 0.3 and 2.7 ± 0.3, respectively, while individuals with the WW genotype gave the least, with 2.3 ± 0.4 and 2.2 ± 0.4 lambs, respectively. Heterozygotes gave intermediary results. 

There is a direct relationship between the number of born and weaned lambs, regardless of the genotype group the animal belongs to. The differences between groups in the number of lambs slaughtered per one ewe for three years indicate an excess of this indicator in heterozygous ewes with the WM genotype compared to WW. Differences were also obtained between animals with MM and WW genotypes. At the same time, no significant difference was found between animals with the MM and WM genotypes in terms of slaughtered lambs per one ewe for three years. From individuals with the MM genotype, more lambs were obtained, but due to the slaughtering of young animals before the time of weaning, their production was close to that in animals with the WM genotype. From an economic point of view, the most profitable yield of lambs comes from animals with MM and WM genotypes compared to WW individuals.

Of the detected alleles, the W allele was more frequent (0.6) than the M (0.4) allele, as a result of low frequency of the homozygotes MM. In spite of a slightly higher level of heterozygous WM than predicted by Hardy–Weinberg equilibrium (0.52 against 0.48), there were no significant deviations (*χ*^2^ = 0.36; df = 1; *p* = 0.84).

Another population of 48 Romanov sheep was investigated to detect the polymorphism within the *BMP-15* gene at 5 mutation points (*FecX^G^* [391 (C-T)], *FecX^H^* [544 (C-T)], *FecX^I^* [579 (T-A)], *FecX^L^* [635 (G-A)], *FecX^B^* [773 (G-T)]) [14]. Among these mutations, the *FecX^G^* locus was the only one to present a polymorphism (Table 4). The frequency of the mutant allele was lower than in the previous population, resulting in an absence of the homozygotes MM. The distribution of genotypes in the *BMPR1B* locus (*Fec^B^*) had a similar pattern as in the *FecX^G^* locus; only homozygous (WW) and heterozygous (WM) genotypes were identified (Table 4). As for the *GDF9* locus, only the wild genotype (WW) was detected.

If the 2 loci are considered together, 13 and 14 heterozygotes were found for *FecX^G^* and *Fec^B^*, respectively, without double heterozygotes. It means than more than a half of this ewe population carry a mutation enhancing the ovulation rate. Moreover, as carriers of both mutations certainly exist, they could have, themselves, a higher ovulation rate than those with either mutation separately [22]. At present, a multiparous merino breed, the Booroola allele, has been first found in Australia [23], a carrier of the mutation *Fec^B^* in the *BMPR1B* gene. Later, this mutation was also found in coarse wool sheep breeds (Hu and small-tailed Han) in China [24,25,26]. The presence of the *BMPR1B* (*Fec^B^*) gene in the studied Tomanov ewes is possibly due to Asian roots of origin. Judging by the works of academician N.I. Vavilov, the flow of animals and consequently genes to the Russian Federation came from China, through Mongolia and Kazakhstan [27]. As for the origin of the *FecX^G^* mutation, it has been found in breeds of very distant countries: Lleyn in North Wales and Small-tailed Han in China, suggesting independent mutation events [28]. This situation does not help to determine the way this mutation appeared in the Romanov breed.

### 3.3. Characteristics of the b Hemoglobin (HB) Locus in Sheep of Various Breeds

In the Romanov breed, a higher frequency of *HB^A^* than the *HB^B^* allele was found, in contrast to most sheep breeds in the world, where the old *HB^B^* allele is predominant (Table 5). The coarse wool, semi-fine or fine wool breeds have a “classical” distribution of alleles and genotypes in the hemoglobin (*HB*) locus. They have a prevalence of *HB^B^* over the *HB^A^* allele, although all the studied breeds have been bred for a long time in rather specific environmental conditions. It should be noted that the HB^A^ allele is a derivative of *HB^B^*, associated with a deletion, i.e., loss of a DNA segment of 40 kb. The high occurrence of the *HB^B^* allele in most of the studied sheep suggests that most breeds were created from populations that carried the “old” *HB^B^* allele [29]. 

We hypothesize that sheep being *HB^A/A^* homozygotes or *HB^A/B^* heterozygotes are more resistant to the conditions of high humidity in Central Russia. For at least one century, the Romanov breed has been reared in a closed regime, without interbreeding with other breeds. A polymorphism at the hemoglobin locus would create opportunities for an easier oxygen attachment by the A variant of the protein, which allows this sheep to survive for a long time in unusual environmental conditions. This situation suggests that gene frequencies for hemoglobin types can serve as a population indicator of extreme conditions caused by different degrees of hypoxia. Establishing the adaptive significance of the studied traits at different levels of biological organization could make it possible to identify a certain role of biochemical polymorphism in the ecological and genetic differentiation of animals [30].

### 3.4. Characterization of the Transferrin (TF) Locus in Sheep of Various Breeds

The function of transferrin is to transport iron ions to reticulocytes, in which hemoglobin biosynthesis takes place. It is the most polymorphic blood protein locus in sheep. The transferrin locus polymorphism was analyzed in 10 sheep breeds of Russia (Romanov, Caucasian, Kuibyshev, Volgogradsk and North Caucasian meat-wool), Azerbaijan (Azerbaijan merino, Karabakh, Bozakh) and Moldova (Tsigay and Karakul) of different productive orientations (Table 6). 

In fine wool and semi-fine wool sheep, the number of genotypes varied between eight and eleven, thus being higher than in the Romanov sheep, with five to six genotypes. In coarse wool sheep, the range is between 5 and 13 genotypes. Seven transferrin alleles were identified—*TF^A^*, *TF^B^*, *TF^C^*, *TF^D^*, *TF^E^*, *TF^P^*, *TF^I^*—the combination of which theoretically gives twenty-eight phenotypes. Specifically, 20 transferrin phenotypes have been identified: *TF^A/A^*, *TF^A/B^*, *TF^A/C^*, *TF^A/D^*, *TF^A/E^*, *TF^A/I^*, *TF^B/B^*, *TF^B/C^*, *TF^B/D^*, *TF^B/E^*, *TF^B/I^*, *TF^C/C^*, *TF^C/D^*, *TF^C/E^*, *TF^C/I^*, *TF^D/D^*, *TF^D/E^*, *TF^D/I^*, *TF^D/P^*, *TF^E/P^*. Phenotypes containing rare alleles have not been detected: *TF^A/P^*, *TF^B/P^*, *TF^C/P^*, *TF^E/I^*, *TF^P/I^*. There were no genotypes homozygous for rare alleles (*TF^E/E^*, *TF^E/P^*, *TF^P/P^* and others). This phenomenon was observed in all three studied populations of Romanov sheep. The most common alleles were *TF^A^*, *TF^B^*, *TF^C^* and *TF^D^*, rarely *TF^E^*, *TF^I^* and *TF^P^*. The *TF^C^* allele had the highest frequency among all the studied breeds (mean = 0.424; min = 0.179; max = 0.656). In the Romanov and coarse wool breeds, the individuals shared a low frequency of the *TF^A^* allele. 

The studied sheep breeds were characterized by a high level of heterozygosity. The Caucasian breed shows the lowest value with 0.48, while the highest one was observed in the third population of the Romanov breed with 0.93. In the populations with a heterozygosity over 0.73, two showed a deviation from genetic equilibrium (the third population of Romanov and Karakul), but not in North-Caucasian meat-wool. In the Caucasian breed, the deviation was mainly due to the absence of BC heterozygotes.

### 3.5. Genetic Structure of the Albumin (ALB) Locus in Sheep of Different Breeds

Albumins are synthesized in the liver and are simple proteins containing up to 600 amino acid residues. They are the lowest molecular weight blood proteins, with values of about 60,000–66,000 D. Their function is to maintain the colloid osmotic pressure of the plasma, the constancy of the concentration of hydrogen ions, and also the transport of various substances. Normally, albumins account for 35–55% of the total amount of blood plasma proteins [31] and can be considered as a certain reserve of amino acids for the synthesis of vital specific proteins in conditions of protein diet deficiency.

The comparison of genetic richness in sheep with different production characters showed that in the coarse wool breeds, including Romanov, the mean number of genotypes was higher than 5.5 in the semi-fine and fine wool breeds, but lower than this value in the semi-fine and fine wool sheep (Table 7). The Volgogradsk breed has the smallest number of genotypes with three variants. The level of heterozygosity in the three populations of the Romanov breed was higher than in the totality of other breeds (ANOVA, *p* < 0.05) with 0.90 compared to 0.758, respectively. The populations showing a significant deviation from the Hardy-Weinberg genetic equilibrium showed heterozygosity values higher or equal to 0.875. Among all the studied breeds, this relates to the three populations of Romanov, the North Caucasian meat-wool and the Volgogradsk breeds.

In semi-fine wool (North Caucasian meat-wool, Tsigay) and fine wool (Azerbaijan merino, Volgogradsk) sheep, the albumin locus showed an absence or rare occurrence of genotypes with the ALB^A^ allele, most likely due to a lower viability of the carriers of these genotypes. There is a great demand for heterozygous genotypes in sheep breeds, regardless of the productive specialization and place of breeding.

### 3.6. Characterization of the Prealbumin (PRE) Locus in Sheep of Different Breeds

In addition to the Romanov breed previously studied [9], the polymorphism of the prealbumin locus (PRE) was investigated in nine breeds of Russia (Caucasian, Volgogradsk, Kuibyshev, North Caucasian meat-wool), Moldova (Tsigay, Karakul) and Azerbaijan (Azerbaijan merino, Karabakh, Bozakh) (Table 8).

In the prealbumin locus, five out of six putative genotypes were identified. The majority of the possible genotypes of this protein were detected among the 10 breeds: *PRE^F/F^*, *PRE^F/S^*, *PRE^F/O^*, *PRE^S/S^* and *PRE^S/O^*. The level of heterozygosity in the studied breeds was above or equal to 0.8 and even reached 1 in each of the 3 populations of Romanov. It should be noted that in all breeds, all phenotypes of the prealbumin protein were found, with the exception of the *PRE^O/O^* variant, which is possibly associated with the non-viability of the offspring. In most of the studied breeds, there was a non-significant deviation from genetic equilibrium in this locus, with the exception of the Caucasian breed. The Romanov breed presented the highest deviation of all the studied breeds.

The comparison of the richness in alleles and genotypes between breeds of different characteristics shows an occurrence of five genotypes and three alleles in fine wool, semi-fine wool and coarse wool breeds. In contrast, there were only three genotypes in the Romanov sheep. As in the albumin locus, the prealbumin locus of the Romanov breed is characterized by an under-representation in homozygous genotypes compared to the other studied breeds, even in the coarse wool group; this may be indicative of a selective advantage of heterozygotic animals.

### 3.7. Comparative Genetic Diversity

An analysis of 20 microsatellite loci on 26 sheep breeds of Eurasia revealed that the Romanov breed was close to the Russian Kulunda settled in the Altai region, and loosely related to 2 Polish breeds (Swiniarka and Wrzosowska) and 4 Scandinavian ones (Swedish Gotland, Gute and Rya, Norwegian Feral) [32]. Using the same set of microsatellites for 24 sheep breeds of Eurasia, Tapio et al. [8] showed a high level of observed heterozygosity in all breeds of the former USSR. We observe a significant positive linear relationship between the sampling size of breeds and the heterozygosity or allelic richness (Figure 1A,B). Within the group of coarse wool sheep of the northwestern cluster cited above, the Romanov breed presented a relatively high degree of heterozygosity and allelic richness. The residuals of the regression between these parameters and population sizes revealed significantly higher values in the group of Romanov–Kulunda–Wrzosowska than in the rest (*t*-test, *p* = 0.0076 and 0.031 for heterozygosity and allelic richness, respectively). It can be concluded that the Romanov belongs to the breeds presenting the highest genetic variability among the northwestern cluster of coarse wool breeds.

Interestingly, Ozerov et al. [33] found that of the five environmental factors investigated, the most important affecting the genetic variability of sheep breeds were a negative effect of geographical latitude and a positive effect of average annual temperature. The sum of their posterior probabilities was 43.1% and 17.8%, which exceeds other factors by almost 9 and 6 times.

### 3.8. Inter-Population Analysis from Seven Blood Group Systems

A total of 576 individuals of the Romanov breed from 5 regions of the Russian Federation were analyzed for 5 genetic indicators of the 7 blood group systems, as follows: level of expected heterozygosity (H_E_); level of observed heterozygosity (H_O_); allelic richness, i.e., number of alleles at the locus (A_R_) averaged over the minimum sample size (n = 38); mean number of alleles per locus (A_M_); coefficient of inbreeding (F) (Table 9).

The analysis showed that the highest level of observed and expected heterozygosity (H_O_) was found in sheep from farm No. 2 of the Voronezh region (0.636 and 0.586, respectively), and the lowest level was found among sheep from farm No. 10 of the Yaroslavl region (0.272 and 0.348, respectively). In the rest of the farms, the level of heterozygosity was intermediate. The maximum level of allelic richness (A_R_) was observed among sheep from the farm of the Republic of Bashkortostan (No. 1: 4.61); the minimum values were detected among sheep from the Smolensk region (No. 4: 2.56). The highest number of alleles per locus (A_M_) was observed in sheep from the Republic of Bashkortostan (No. 1) and farm No. 3 from the Voronezh region (4.71), and the minimum among sheep of farm No. 4 of the Smolensk region. The coefficients of inbreeding (f) did not significantly deviate from zero in any of the studied populations.

A cluster analysis between these 12 populations showed the proximity of most of the studied groups, due to the fact that livestock on farms were actually formed from breeding reproducers of the Yaroslavl region (Figure 2A,B); the farms 6, 7, 8, 9 and 12 were actively involved in the export of animals, in contrast to the farms 10 and 11, which appeared to be isolated. This does not seem to be the case for farms 3 and 4 of Voronezh and Smolensk, respectively. In the city of Yaroslavl, the association created for Romanov sheep at the Joint Stock Company “Yaroslavskoye” for breeding activity preserves the genetic diversity in the breed. For this, the lineages of outstanding rams are clearly traced in the farms. Specialists try to maintain high maternal qualities so that each lamb can receive their dam’s milk. Under unforeseen circumstances, even dairy cows can provide milk to the lambs.

## 4. Conclusions

Based on the new results presented here, we provide new knowledge of different gene pools of the Romanov breed. Our analyses clarified the description of the blood group systems A, B, C and M in comparison to the previous work published in 2015 [9]. The identification of antigens of blood group systems in eight species of ruminants was carried out. The greatest number of common antigens was determined between sheep and goat on one hand, and between cattle and yak on the other. The lowest number of analogues were found in the triad sheep–goats–cattle.

A high ovulation rate is one of the important features of Romanov sheep. A description of the polymorphism of the *BMP-15* locus taken from a paper published in 2019 in the Russian language was given [15], but updated analyses on a new sample of animals were conducted. The mutations at *FecX^G^* (*BMP-15* gene) and *Fec^B^* (*BMPR1B* gene) were identified for the first time in the Romanov breed. Regarding the hemoglobin locus, the previous results obtained for the Romanov breed in the paper of 2015 [9] were extended to nine breeds with coarse, semi-fine and fine wool breeds of the Russian Federation, allowing a comparative analysis. The same analysis was conducted with the transferrin, albumin and prealbumin loci with the same set of sheep breeds, again placing the Romanov breed in a larger context.

The history of Romanov sheep is poorly understood; there are several hypotheses, although the first mentions date back to the 19th century [33]. It is thus one of the oldest breeds in Russia and has been bred for a long time without crossing with other breeds. These new data allowed us to address the question of the enigmatic origin of the Romanov breed. From the studies of a set of 20 microsatellites [8,32], it has been concluded that this breed takes a particular position among the coarse wool breeds. Tapio et al. [8] first highlighted some kinship with a fat-tailed group of “native breeds from the Caucasus and steppes of the Caspian basin and Kazakhstan”. On the other hand, Ozerov et al. [32] pointed out a very basal branching in the “northwestern cluster involving the breeds from Sweden, Norway, and Poland”.

The connection between the Romanov and coarse wool breeds is supported by a weak piece of evidence: a common low frequency in the allele *TF^A^* of the transferrin locus and a more frequent allele *PRE^O^* in the prealbumin locus. Regarding the originality of the Romanov breed, the pattern of alleles within the different blood systems seems to be unconnected to the three Ukrainian breeds, but this result is preliminary given the lack of homogeneity in the accuracy of analyses. The prevalence of the b *HB^A^* allele in the hemoglobin locus is unique among the coarse, semi-fine and fine wool analyzed breeds, for which the allele *HB^B^* prevails. In the transferrin locus, the *TF^C^* allele is dominant in all the breeds investigated, but the Romanov breed shows the highest frequency. In the albumin locus, the Romanov sheep differed from the others due to its higher frequency of allele *ALB^A^* and lower frequency of *ALB^B^*. In the prealbumin locus, the Romanov is the only breed without carriers of homozygous animals FF and SS. Taken together, these features suggest that the Romanov sheep experienced a genetic isolation from other breeds for a long period. Unexpectedly, the discovery of the *Fec^B^* mutant (Booroola) in the *BMPR1B* gene in the Romanov sheep raises the question of its appearance, probably dating back several centuries. Having arisen on the Hindustan Peninsula, the mutation, by chance, ends up in Australia, where a new breed was being formed [9,23]. Later, European, and then New Zealand, Chinese and African scientists identified in their breeds several genes responsible for a high ovulation rate [21,24,25,34].

Regarding the adaptation of the Romanov sheep to the harsh conditions in the central regions of the Russian Federation, several traits of its fur have been already mentioned. Our results indicated the high degree of heterozygosity of this breed for several loci. First of all, only heterozygotes were found in the prealbumin locus, and this was nearly also the case for the albumin locus in two populations among the three investigated, in contrast to other breeds. Moreover, the data based on microsatellites emphasized the genetic variability of the Romanov sheep among the northwestern cluster of coarse wool breeds. The reason for interest in the heterozygosity recorded in the populations of Romanov sheep is that it could lead to a higher range of activity of the proteins coded by the different alleles. However, the resistance of this breed could also be due to advantageous variants, as hemoglobin is encoded by the A allele.

From a genetic point of view, the following measures are currently in place or will be adopted to maintain the Romanov breed’s genetic variability in several farms within the Yaroslav region: rotation of breeding sheep in order to avoid inbreeding; mandatory testing of the breeding core for genetic markers; creation of a sperm bank from elite rams and embryos from ewes.

## Figures and Tables

**Figure 1 animals-13-01320-f001:**
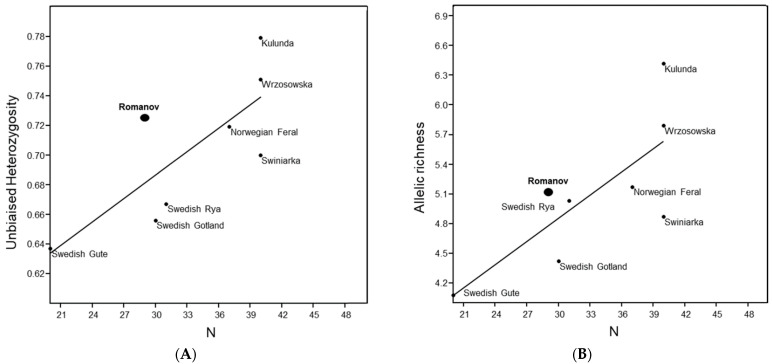
Genetic variability of sheep in the northwestern cluster (northwest of Russia, Poland and Scandinavia), reported to the sampling size. (**A**): Unbiased heterozygosity; (**B**): allelic richness. N = sampling size. The *p*-values of regressions are 0.024 and 0.028 for A and B, respectively.

**Figure 2 animals-13-01320-f002:**
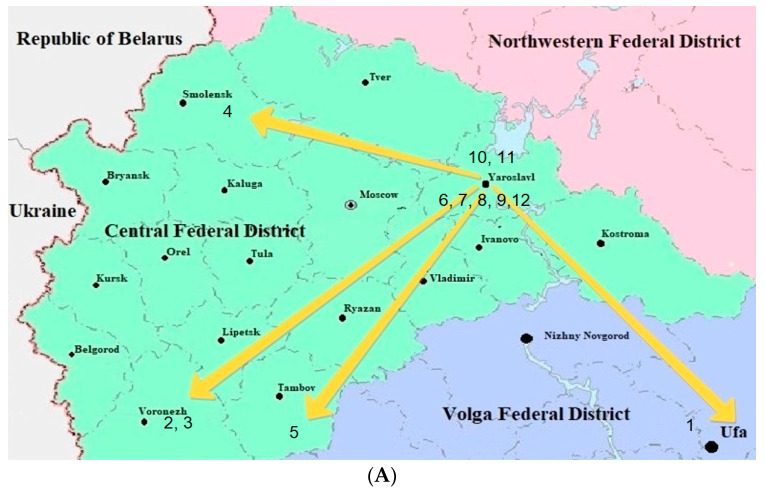
Relationships between 12 Romanov sheep populations from different regions of the Russian Federation. (**A**): geographical map. The arrows indicate possible directions of animal export. (**B**): phylogenetic relationship based on the cluster analysis. Bootstrap values are indicated at each node.

**Table 1 animals-13-01320-t001:** Frequency of genetic factors of 6 systems of blood groups in the Romanov breed.

System	Genotype	Frequency Occurrence	Allele	Frequency Occurrence	Phenotypic Classes	Frequency Occurrence
A (n = 52)	*A^-/-^*	0.3654	*A^-^*	0.5884	Aa	0.1346
	*A^b/b^*	0.0798	*A^b^*	0.2899	Aab	0.0962
	*A^a/a^*	0.0126	*A^a^*	0.1217	Ab	0.4038
	*A^a/b^*	0.0962			A-	0.3654
	*A^b/-^*	0.324				
	*A^a/-^*	0.1220				
C (n = 52)	*C^-/-^*	0.3462	*C^-^*	0.5827	C-	0.3462
	*C^b/b^*	0.0918	*C^b^*	0.3055	Cb	0.4423
	*C^a/a^*	0.0118	*C^a^*	0.1118	Cab	0.0769
	*C^a/b^*	0.0769			Ca	0.1346
	*C^b/-^*	0.3504				
	*C^a/-^*	0.1229				
M (n = 52)	*M^-/-^*	0.0192	*M^-^*	0.0734	M-	0.0192
	*M^c/c^*	0.1225	*M^a^*	0.0481	Mac	0.0962
	*M^c/-^*	0.1082	*M^c^*	0.5516	Mc	0.2308
	*M^a/c^*	0.0963	*M^b^*	0.2596	Mbc	0.5192
	*M^b/c^*	0.5192	*M^ab^*	0.0673	Mabc	0.1346
	*M^ab/c^*	0.1346				
D (n = 52)	*D^a/a^*	0.0192	*D^a^*	0.1340	Da	0.2500
	*D^-/-^*	0.7500	*D^-^*	0.8660	D-	0.7500
	*D^a/-^*	0.2308				
R (n = 37)	*R^R/R^*	0.0541	*R*	0.2466	R	0.4324
	*R^R/r^*	0.3784	*r*	0.7534	O	0.5676
	*R^r/r^*	0.5675				
I (R = 52)	*I^I/I^*	0.2115	*I*	0.4629	I	0.7115
	*I^I/i^*	0.5000	*i*	0.5371	i	0.2885
	*I^i/i^*	0.2885				

Note: *A^-/-^* designs a genotype with two null alleles (*A^-^*), *A^a/b^* a heterozygote with *A^a^* and *A^b^* alleles.

**Table 2 animals-13-01320-t002:** Frequency of genetic factors in the B blood group system in the Romanov breed (n = 52).

System	Genotype	Frequency	Allele	Frequency	Phenotypic Classes	Frequency
		Occurrence		Occurrence		Occurrence
	*B^-/-^*	0.099	*B^-^*	0.2533	Bbdg	0.1923
	*B^b/-^*	0.0186	*B^b^*	0.0412	Bbeig	0.0769
	*B^g/g^*	0.0007	*B^d^*	0.0585	Bdg	0.0769
	*B^b/g^*	0.0385	*B^e^*	0.0075	B-	0.1154
	*B^d/g^*	0.0769	*B^i^*	0.0249	Bdeg	0.0576
	*B^bdg/-^*	0.1923	*B^g^*	0.0928	Bbdig	0.2885
	*B^bdg/b^*	0.0107	*B^bde^*	0.0033	Bdeig	0.0192
B	*B^bdg/d^*	0.0178	*B^bdi^*	0.056	Bg	0.0192
	*B^bdg/g^*	0.0113	*B^bdg^*	0.1808	Bbdeig	0.0385
	*B^bdg/bdg^*	0.009	*B^bei^*	0.0405	Bbei	0.0385
	*B^bei/-^*	0.0306	*B^beg^*	0.0179	Bbg	0.0385
	*B^bei/b^*	0.0022	*B^big^*	0.0961	Bbig	0.0385
	*B^bei/e^*	0.0022	*B^dei^*	0.0078		
	*B^bei/i^*	0.0022	*B^deg^*	0.0379		
	*B^bei/bei^*	0.0012	*B^dig^*	0.0591		
	*B^big/-^*	0.0305	*B^eig^*	0.0224		
	*B^big/b^*	0.0022				
	*B^big/i^*	0.0022				
	*B^big/g^*	0.0023				
	*B^big/big^*	0.0012				
	*B^deg/-^*	0.0457				
	*B^deg/d^*	0.0033				
	*B^deg/e^*	0.0033				
	*B^deg/g^*	0.0034				
	*B^deg/deg^*	0.0019				
	*B^bdi/g^*	0.0037				
	*B^bdg/i^*	0.0363				
	*B^big/d^*	0.0242				
	*B^dig/b^*	0.0218				
	*B^bdi/bdg^*	0.0363				
	*B^bdi/big^*	0.0242				
	*B^bdg/eig^*	0.0403				
	*B^bdi/dig^*	0.0218				
	*B^bdg/dig^*	0.0363				
	*B^big/dig^*	0.0242				
	*B^bei/g^*	0.0032				
	*B^beg/i^*	0.007				
	*B^big/e^*	0.0078				
	*B^eig/b^*	0.007				
	*B^bei/beg^*	0.0078				
	*B^bei/big^*	0.0086				
	*B^beg/big^*	0.0078				
	*B^bei/eig^*	0.0078				
	*B^beg/big^*	0.007				
	*B^big/eig^*	0.0078				
	*B^dei/g^*	0.0018				
	*B^deg/I^*	0.002				
	*B^dig/e^*	0.0018				
	*B^eig/d^*	0.0018				
	*B^dei/deg^*	0.002				
	*B^dei/dig^*	0.0018				
	*B^deg/dig^*	0.002				
	*B^dei/eig^*	0.0018				
	*B^deg/eig^*	0.002				
	*B^dig/eig^*	0.0018				
	*B^bde/big^*	0.0023				
	*B^bdi/beg^*	0.0021				
	*B^bdg/bei^*	0.0038				
	*B^bdi/dig^*	0.0021				
	*B^bdi/deg^*	0.0025				
	*B^bdg/dei^*	0.0035				
	*B^bde/eig^*	0.0021				
	*B^bei/deg^*	0.0027				
	*B^beg/dei^*	0.0021				
	*B^bdi/eig^*	0.0021				
	*B^bei/dig^*	0.0023				
	*B^big/dei^*	0.0023				
	*B^bdg/eig^*	0.0035				
	*B^beg/dig^*	0.0021				
	*B^big/deg^*	0.0027				

Note: *B^-/-^* designs a genotype with two null alleles*, B^g/g^* a homozygote for allele *B^g^*, and *B^d/g^* a heterozygote for alleles *B^d^* and *B^g^*.

**Table 3 animals-13-01320-t003:** Characteristics of ewes with different genotypes of the *BMP-15* locus. Out of 52 animals studied, 2 were sires.

Indicators	Genotype at the *BMP-15* Locus		
	WW	WM	MM
Number of identified ewes with genotypes (n = 50)	17	26	7
Frequency of genotypes	0.34	0.52	0.14
Number of barren ewes in 3 years (%)	4 (23.5%)	5 (19.2%)	1 (14.2%)
The number of non-productive ewes for 3 years (%)	4 (23.5%)	3 (11.2%)	0 (0%)
Number of productive ewes (n = 43)	13	23	7
Obtained lambs for 3 years	92	181	63
Weaning lambs for 3 years	85	170	56
Obtained lambs per ewe on average per year	2.3 ± 0.4	2.6 ± 0.4	3.0 ± 0.3
Weaning lambs per ewe on average per year	2.2 ± 0.4	2.5 ± 0.4	2.7 ± 0.3

**Table 4 animals-13-01320-t004:** Characteristics of the *BMP-15* (*FecX^G^*) *and BMPR1B* (*Fec^B^*) loci in Romanov ewes (n = 48).

			Genotype			Allele Frequency				
Locus	Ewe Numbers		WW	WM	MM	W	M	*χ* ^2^	*Df*	*p*
*BMP-15*	48	Obs	35	13	0	0.865	0.135	1.18	1	0.6
*FecX^G^*		Exp	35.88	11.24	0.88					
*BMPR1B*	48	Obs	34	14	0	0.854	0.146	1.4	1	0.497
*Fec^B^*		Exp	35.02	11.96	1.02					

Note: Obs: observed number of genotypes; Exp: expected number of genotypes; W—Wild, wild or normal allele; M—Mutant, mutant allele; WW, homozygous genotype for the wild or normal allele; WM, heterozygous genotype; MM, homozygous genotype for the mutant allele. The conformity to Hardy–Weinberg equilibrium was assessed through a *χ*^2^ test.

**Table 5 animals-13-01320-t005:** Comparative analysis of the state of the hemoglobin locus (*HB*) in sheep of various breeds in Russia, Azerbaijan and Moldova (n = 497). Conformity to Hardy–Weinberg was tested by *χ*^2^ tests.

Breeds	Occurrence of Genotypes at the Hemoglobin Locus			Allele Frequency		*χ* ^2^	*Df*	*p*
	AA	AB	BB	A	B			
Populations of sheep of the Romanov breed from various regions of Russia								
I, n = 50	24	19	7	0.67	0.33	1	1	>0.05
II, n = 47	25	17	5	0.713	0.287	0.64	1	>0.05
III, n = 40	14	20	6	0.6	0.4	0.168	1	>0.05
Coarse wool breeds								
Bozakh, n = 40	2	4	34	0.1	0.9	7.9	1	<0.01 **
Karabakh, n = 40	0	0	40	0	1	0	1	>0.05
Karakul, n = 40	3	12	25	0.225	0.775	0.782	1	>0.05
Semi-fine wool breeds								
Kuibyshev, n = 40	6	10	24	0.275	0.725	5.55	1	<0.05 *
North Caucasian meat-wool, n = 40	2	11	27	0.188	0.812	0.39	1	>0.05
Tsigay, n = 40	1	7	32	0.1125	0.8875	0.602	1	>0.05
Fine wool breeds								
Azerbaijan merino, n = 40	2	3	35	0.0875	0.9125	11.09	1	<0.001 ***
Volgogradsk, n = 40	5	18	17	0.35	0.65	0.005	1	>0.05
Caucasian, n = 40	1	18	21	0.25	0.75	1.6	1	>0.05

Note: *—*p* < 0.05; **—*p* < 0.01; ***—*p* < 0.001.

**Table 6 animals-13-01320-t006:** Occurrence of alleles and the status of genetic equilibrium in the transferrin (*TF*) locus in sheep of various breeds (n = 497). Fisher’s exact tests were performed on the genotype occurrences for each population (see Table 4 in Marzanov et al. [9]). He = heterozygosity.

Breeds	Occurrence of Alleles at the Transferrin Locus									
	*TF^A^*	*TF^B^*	*TF^C^*	*TF^D^*	*TF^E^*	*TF^I^*	*TF^P^*	*Df*	*p*	He
Populations of sheep of the Romanov breed from various regions of Russia										
I, n = 46	0.033	0.348	0.587	0.033				12	0.452 NS	0.65
II, n = 48	0.063	0.25	0.656	0.031				12	0.374 NS	0.63
III, n = 46	0.120	0.424	0.38	0.08				12	0.0001 ***	0.93
Coarse wool breeds										
Bozakh, n = 40	0.075	0.225	0.475	0.2			0.025	12	0.513 NS	0.70
Karabakh, n = 40	0.113	0.237	0.325	0.237	0.088			17	0.45 NS	0.70
Karakul, n = 40	0.138	0.475	0.275	0.11				12	0.0092 **	0.73
Semi-fine wool breeds										
Kuibyshev, n = 39	0.4231	0.1026	0.321	0.013	0.01			20	0.446 NS	0.74
North Caucasian meat-wool, n = 40	0.225	0.2375	0.425	0.063	0.025	0.013	0.013	21	0.607 NS	0.80
Tsigay, n = 39	0.2308	0.1538	0.564	0.0513				21	0.459 NS	0.51
Fine wool breeds										
Azerbaijan merino, n = 40	0.063	0.300	0.413	0.013	0.025	0.063	0.012	21	0.087 NS	0.65
Volgogradsk, n = 39	0.3333	0.359	0.179	0.090		0.013	0.003	16	0.098 NS	0.56
Caucasian, n = 40	0.275	0.188	0.488	0.013	0.025	0.013		21	0.012 *	0.48

Note: *—*p* < 0.05; **—*p* < 0.01; ***—*p* < 0.001.

**Table 7 animals-13-01320-t007:** Polymorphism of the albumin (*ALB*) locus in various breeds of sheep (n = 504).

Breeds	Albumin Locus Genotypes										Allele Frequency				He	*Df*	*p*
	AA	AB	AC	AD	BB	BC	BD	CC	CD	DD	A	B	C	D			
Populations of the Romanov sheep from various regions of Russia																	
I, n = 46		3		2		20	12		9		0.055	0.38	0.315	0.25	1	8	0.009 **
II, n = 48		4			1	16	12		15		0.042	0.354	0.323	0.28	0.979	8	0.006 **
III, n = 50	2	18	9		1	8		1	1		0.31	0.28	0.4	0.01	0.72	8	0.036 *
Coarse wool breeds																	
Bozakh, n = 40		4	14	8		1		6	6	1	0.325	0.063	0.412	0.2	0.825	8	0.403 NS
Karabakh, n = 40						4	9	4	12	11	0	0.163	0.3	0.537	0.625	4	0.994 NS
Karakul, n = 40		1	1		12	19	2	4		1	0.025	0.575	0.35	0.05	0.575	8	0.992 NS
Semi-fine wool breeds																	
Kuibyshev, n = 40		1	1		6	20	6	5	1		0.025	0.488	0.4	0.087	0.725	8	0.700 NS
North Caucasian meat-wool, n = 40					2	15	9	11		3	0	0.35	0.325	0.325	0.6	4	0.012 *
Tsigay, n = 40					6	30	1	3			0	0.538	0.45	0.012	0.775	4	0.046 *
Fine-wool breeds																	
Azerbaijan merino, n = 40					6	13	19	1	1		0	0.55	0.2	0.25	0.825	4	0.084 NS
Volgogradsk, n = 40					5	15	20				0	0.5625	0.1875	0.25	0.875	4	0.009 **
Caucasian, n = 40		1	2		6	19	2	10			0.038	0.425	0.512	0.025	0.6	8	0.893 NS

Note: *Df* : degrees of freedom for the Fisher’s exact test about genetic equilibrium; He = heterozygosity; NS: non-significant, *: *p* < 0.05, **: *p* < 0.01.

**Table 8 animals-13-01320-t008:** Prealbumin polymorphism (*PRE*) in different breeds (n = 504).

Breeds	Prealbumin Locus Genotypes						Allele Frequency				*Df*	*p*
	FF	FS	FO	SS	SO	OO	F	S	O	He		
Populations of sheep of the Romanov breed from various regions of Russia												
I, n = 46		19	6		21		0.272	0.435	0.293	1	4	0.0011 **
II, n = 48		11	7		30		0.188	0.427	0.385	1	4	0.0032 **
III, n = 50		13	29		8		0.42	0.21	0.37	1	4	0.0025 **
Coarse wool breeds												
Bozakh, n = 40	1	20	6	4	9		0.35	0.463	0.188	0.88	4	0.25 NS
Karabakh, n = 40	1	23	7	3	6		0.4	0.438	0.163	0.90	4	0.11 NS
Karakul, n = 40	1	20	10	1	8		0.4	0.375	0.225	0.95	4	0.07 NS
Semi-fine wool breeds												
Kuibyshev, n = 40	4	26	4	3	3		0.475	0.438	0.088	0.83	4	0.19 NS
North Caucasian meat-wool, n = 40	6	15	14	2	3		0.513	0.275	0.213	0.80	4	0.49 NS
Tsigay, n = 40	5	23	4	3	5		0.463	0.425	0.113	0.80	4	0.63 NS
Fine-wool breeds												
Azerbaijan merino, n = 40	4	19	12	1	4		0.488	0.313	0.2	0.88	4	0.23 NS
Volgogradsk, n = 40	5	17	15	1	2		0.525	0.263	0.213	0.85	4	0.15 NS
Caucasian, n = 40	1	21	14	2	2		0.463	0.338	0.2	0.93	4	0.014 *

Note: Deviation from genetic equilibrium was deduced from Fisher’s exact test. *—*p* < 0.05; **—*p* < 0.01, NS: non-significant; He = heterozygosity.

**Table 9 animals-13-01320-t009:** Characteristics of sheep of the Romanov breed from various regions of the Russian Federation (n = 576).

Number Farms	n	H_O_	H_E_	A_R_	A_M_	F
1	47	0.52	0.581	4.61	4.71	0.107
2	40	0.636	0.585	4.14	4.14	−0.087
3	52	0.335	0.561	4.66	4.71	0.405
4	50	0.44	0.424	2.56	2.57	−0.039
5	50	0.574	0.569	3.96	4.00	−0.01
6	40	0.614	0.579	4.27	4.29	−0.061
7	39	0.553	0.571	4.14	4.14	0.032
8	40	0.596	0.526	4.11	4.14	−0.135
9	40	0.579	0.558	4.41	4.43	−0.037
10	50	0.272	0.348	2.85	2.86	0.221
11	50	0.411	0.412	2.71	2.71	0.001
12	38	0.541	0.548	3.86	3.86	0.012

## Data Availability

Some of the research results obtained were published in open access, in academic journals of the Russian Federation (see [4,5,6,7,8,9,14]). Some materials are presented for the first time in this article.

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
