# Peer review of "The Significance of a Multilocus Analysis for Assessing the Biodiversity of the Romanov Sheep Breed in a Comparative Aspect"

_animals, 2023, doi:10.3390/ani13081320_

Round 1

Reviewer 1 Report (Previous Reviewer 2)

Dear authors,

It is evident that you have made substantial progress in the paper you changed when compared to the original form. Yet I believe that some fundamental principles are being missed:

One is that the last phrase of the Simple summary is completely unrelated to what you really did. Simply provide the outcomes achieved in response to the tasks assigned to you.

A conclusion that appears to be a continuation of the second Discussion is extremely tiresome for the reader. It would be useful if you relocated the information you have provided here to the previous section and merely presented the core of what you have done here.

Due to the explanations you provide in the table titles, it is difficult to focus on them. Why don't you give them a footnote and attempt to bring attention to the title?

L617: Please indicate which “sheep breeds”.

L619: Please provide several references for “Vavilov”.

Author Response

Please find attached the responses

Reviewer 2 Report (Previous Reviewer 1)

The paper is improved much after corrections. The results are important for specialists. 

Below some detaled comments:

Table 1.

For C group the sum of genotype frequencies is not equal 1

Table 2

The results of genotype frequencies are impossible to obtain. The sample size is 52 animals, so the lowest frequency of genotype could be 1/52 = 0.0192

Line 398

muskox should one word

Table 4

Number of animals is 50 and 2 out of them are sires. In table there is number of productive ewes 43 and 7 nonproductive ewes, together 50. Where are sires?

In first line is star (n=50)*, which is not explained below

Line 523-525

The explanation is not clear. Lamb slaughtered when is weaned should counted as weaned, so the difference between born and weaned are because of mortality in the period between.

Line 612

What the reproduction results do heterozygotes carriers of  FecXG or FecB mutations ?

Table 6

Why degree of freedom of Chi-square test is equal to 1 – should be 2 because the observed and expected genotypes frequencies are compared.

Table 8.

For such small number of animals and large degree of freedom the Chi-square test is not valid – the condition of the test is not less than 5 expected number in the one group. There are not possibility to check statistically the distribution in this situation – low N and large df

P value for Karakul must be corrected

Table 9

Why df = 3 ?

Is the He calculated properly? For example Bazakh breed – there is 40 animals: 5 homozygotes and 35 heterozygotes, so He=35/40 = 0.875 but in the table is 0.98 – the same is for all breeds except Romanov

Line 835

In text is ”f” is and in table “F” – should be the same

Author Response

Please find attached the responses

This manuscript is a resubmission of an earlier submission. The following is a list of the peer review reports and author responses from that submission.

Round 1

Reviewer 1 Report

The paper is interesting but must be rearanged to be readable. Too many results presented without proper statistical analysis. Some results could be presented much better. Some results seems to be of other authors or explanation is not clear. The English is very strange because not proper words are used. 

Some detailed comments are below but there could be much more. 

11.       In table 1 – Please check the values of genes frequencies because it seems to be mistaken – f.e.  the homozygote genotype AbAb frequency is twice larger than of allele Ab.

22.       The text from part “Results and discussion” (lines 148-156) should be moved to “Material and methods”

33.       Lines 171-172 – the statistical test is needed for this comparison

44.       Lines 219 and others – I am not sure that comparison between mosaic proportion between so different in the point of reproduction species as sheep and cattle are is reasonable. This part does not fit very well to description of Romanow breed and could be excluded

55.       Lines 251-252 – the used description means  heterozygosity ?

66.       Table 4 – what means retired eves or dry eves or counted lambs ewes?, why number and percentage is in different lines? – this table must rearranged and it is not needed to repeat the same data. Also the sum of eves with and without lams are not the same as total number

77.       Lines 263-264 – numbers are not connected with table 4. The total number of lambs without number of mothers are not possible to compare. There is lack of statistical comparison between results

88.       Line 269 what means: ”number of lambs killed per 1 ewe” or “of slaughtered lambs per 1 ewe” or ”the departure of young animals at the time of weaning” – are those words mean just weaned lamb?

99.    Table 5 – the frequency od the same locus is in table 4 but values are different? What was compared using the Chi-square test – if genotypes why df is equal to 1 (in all table is the same df)

  10.  Tables 5 and 6 could put together

111.   Line338 – what means Romanov sheep was “bred "in itself", - not proper words are used

112.   Lines 338 vs 314  in one there was not influence of other breeds in other place there is information about Chinese breed influence

113.   The degree of freedom have very strange values in table 7 and 8 – what test is used?

114.   Lines f.e. 373-378 presenting values from table without any comments can be excluded

115.   Generally – if the results are used in the text will be good to include  table as reference

116.   Lines 471 and 472 – mistakes in names of breeds

117.   Figure 1 - the figure is taken from cited papers or prepared by authors? What is statistical value of such comparison?

118.   Lines 488-491 – this is part of material and methods or cited data?

119.   Table 11 – if herds are from different geographic locations it could be more useful to show results on the map – the names of regions of Russia means almost nothing to regular reader. Are difference between herds significant or not. The figure 2 is not possible to understand. What was exchange of genetic material between herds? If herds were in small geographic distance such exchange is more probable. So comparison of herds within one breed could be influenced by breeding program.

Author Response

The paper is interesting but must be rearranged to be readable. Too many results presented without proper statistical analysis. Some results could be presented much better. Some results seem to be of other authors or explanation is not clear. The English is very strange because not proper words are used. 

The statistics were entirely revised and many paragraphs were re-written. We explained in detail in material and method section what were the parts taken from published papers and what were the novelties. This was reminded in the beginning of the conclusion.

Some detailed comments are below but there could be much more. 

  1. In table 1 – Please check the values of genes frequencies because it seems to be mistaken – f.e.  the homozygote genotype AbAbfrequency is twice larger than of allele Ab.

Done

  1. The text from part “Results and discussion” (lines 148-156) should be moved to “Material and methods”

Done

  1. Lines 171-172 – the statistical test is needed for this comparison

Done. In line 190: It indicates that among the sheep breeds considered, Romanov ewes showed the highest birth rate of monozygotes (χ2 test, p=0.0095).

  1. Lines 219 and others – I am not sure that comparison between mosaic proportion between so different in the point of reproduction species as sheep and cattle are is reasonable. This part does not fit very well to description of Romanow breed and could be excluded

This part has been reorganized and almost half of the section about cattle has been removed.

  1. Lines 251-252 – the used description means heterozygosity?

The paragraph has been reorganized as two populations of Romanov sheep were considered. A first one already published and concern the table 4 and its comments, and a new one with the precise determination of the mutation (table 5).

  1. Table 4 – what means retired eves or dry eves or counted lambs ewes? why number and percentage is in different lines? – this table must rearranged and it is not needed to repeat the same data. Also the sum of eves with and without lams are not the same as total number

The mistakes came from a wrong translation of the Russian text. The new version provides the necessary corrections.

  1. Lines 263-264 – numbers are not connected with table 4. The total number of lambs without number of mothers are not possible to compare. There is lack of statistical comparison between results

See previous remarks.

  1. Line 269 what means: ”number of lambs killed per 1 ewe” or “of slaughtered lambs per 1 ewe” or ”the departure of young animals at the time of weaning” – are those words mean just weaned lamb?

See previous remarks

  1.   Table 5 – the frequency of the same locus is in table 4 but values are different? What was compared using the Chi-square test – if genotypes why dfis equal to 1 (in all table is the same df)

As written above, these results come from a different population.

  1. Tables 5 and 6 could put together

Done

  1. Line338 – what means Romanov sheep was “bred "in itself", - not proper words are used

In line 325, we corrected the sentence: For at least one century, the Romanov breed has been reared in a closed regime, without the influence of someone else's blood.

  1. Lines 338 vs 314 in one there was not influence of other breeds in other place there is information about Chinese breed influence

You are right, this point is discussed in the conclusion section. It is true that this breed was genetically isolated since at least 150 years. But before, the documentation is absent. In line 529, we wrote:

Unexpectedly, the discovery of FecB mutant (Booroola) in the BMPR1B gene in the Romanov sheep raises the question of its appearance, probably dating back several centuries.

  1. The degree of freedom have very strange values in table 7 and 8 – what test is used?

Always Chi square tests, but all was recalculated.

  1. Lines f.e. 373-378 presenting values from table without any comments can be excluded

Removed

  1. Generally – if the results are used in the text will be good to include table as reference

Thank you, done

  1. Lines 471 and 472 – mistakes in names of breeds

Done

  1. Figure 1 - the figure is taken from cited papers or prepared by authors? What is statistical value of such comparison?

It is new, and we add some statistics to point out the high degree of genetic diversity of Romanov breed.

  1. Lines 488-491 – this is part of material and methods or cited data?

We took the data from the paper of Tapio et al (supplementary data) and used them to make calculations and draw the figure 1. We added a small part at the end of the material and method section, line 133: To compare the heterozygosity and allelic richness obtained from 20 microsatellite markers in the group of coarse wool sheep (including the Romanov breed) of the northwestern cluster of the former URSS (Tapio et al. [47]), we calculated the residuals of these parameters against population sizes. The residuals of the group including the Romanov breed was themselves compared to the rest using a t-test. 

  1. Table 11 – if herds are from different geographic locations it could be more useful to show results on the map – the names of regions of Russia means almost nothing to regular reader. Are difference between herds significant or not. The figure 2 is not possible to understand. What was exchange of genetic material between herds? If herds were in small geographic distance such exchange is more probable. So, comparison of herds within one breed could be influenced by breeding program.

Thank you for the suggestion. A map was added and the comments, accordingly.

Reviewer 2 Report

In this study, the objective was to determine the genetic characteristics of Romanov sheep. Although the introduction is sufficient, it is recommended that the material method, abstract, and conclusion be reviewed once again. Furthermore, in my opinion, this manuscript was written as an article rather than a review. As a result, most of the uncertainties are caused by this type of writing.

Best regards,

Major Comments

Material and method part is unclear, as well as abstract section.

“Gene diagnostics” should be removed from the title. This term used for medical field. “A comparative assessment of the efficiency of its use in assessing the biodiversity of the Romanov sheep breed” is recommended.

There are a lot of references in the conclusion section, which needs to be shortened, clarified, and generalized. This section should be rearranged to make the results of the study more clear.

L494: One more column should be added to Table 11 and given in each region.

The tables should be presented in the standard format.

There is no indication as to whether the data are from your own study or from another, particularly in tables comparing species.

Minor Comments

References should be written according to writing guide.

In line 215, “chi square” or “χ2” should be written instead of “Chi2”

N numbers of each genotype should be given in table 1 and 2.

Observation values provided in Table 3 are unclear as to how and where they were obtained.

In line 165, It is given in the table that there are 181 Romanov lambs given as 182 in the table. Whichever is correct should be corrected.

In line 532, “Romanov breed” or “Romanov Sheep” should be written instead of “Sheep of the Romanov breed”

L478, L508, L511: Please use “Figure” instead of “Fig.”.

The explanations provided in the table titles should be provided as footnotes.

Author Response

In this study, the objective was to determine the genetic characteristics of Romanov sheep. Although the introduction is sufficient, it is recommended that the material method, abstract, and conclusion be reviewed once again. Furthermore, in my opinion, this manuscript was written as an article rather than a review. As a result, most of the uncertainties are caused by this type of writing.

Best regards,

Major Comments

Material and method part is unclear, as well as abstract section.

We perfectly agree with your remark. As there is a complex relationship between already published and novelties, we indicated carefully on the material and method section what was new and what was taken from published works. 

“Gene diagnostics” should be removed from the title. This term used for medical field. “A comparative assessment of the efficiency of its use in assessing the biodiversity of the Romanov sheep breed” is recommended.

Done

There are a lot of references in the conclusion section, which needs to be shortened, clarified, and generalized. This section should be rearranged to make the results of the study more clear.

We re-wrote the conclusion in structured paragraphs and we hope that the present version s more satisfactory.

L494: One more column should be added to Table 11 and given in each region.

Following the suggestion of reviewer 1, we added a map indicating the location of each farm, and added the comments accordingly.

The tables should be presented in the standard format.

Many tables have been changed.

There is no indication as to whether the data are from your own study or from another, particularly in tables comparing species.

 Done

Minor Comments

References should be written according to writing guide.

Done, as far as we know.

In line 215, “chi square” or “χ2” should be written instead of “Chi2”

Done

N numbers of each genotype should be given in table 1 and 2.

We indicated for each system the sampling size of the population, so the number of genotypes become redundant. 

Observation values provided in Table 3 are unclear as to how and where they were obtained.

It is now indicated in material and method section. The data concerning the Romanov breed were completed with 79 new animals analyzed. For the rest (other sheep breeds, goats and cattle, the data were taken from the publication of 2019.

In line 165, It is given in the table that there are 181 Romanov lambs given as 182 in the table. Whichever is correct should be corrected.

Done

In line 532, “Romanov breed” or “Romanov Sheep” should be written instead of “Sheep of the Romanov breed”

Done

L478, L508, L511: Please use “Figure” instead of “Fig.”.

Done

The explanations provided in the table titles should be provided as footnotes.

We think it is more easy to read when the explanations are included in the legend of the table.

Round 2

Reviewer 1 Report

This paper is very difficult to read. The words used are not proper for the subject and this make reading really hard. Also there are too many results, in some part is no information about samples size, on the end is information about microsatellites not mention in Material and method. In tables there is not needed to include duplicated information, like frequency of phenotypes when frequency of genotypes is presented. The X-square test - strange is so large degree of freedom. Also information about analyzed breeds are doubled bit for other is too small data. On the dendrogram are many compared populations (farms) and there is no information about them. What was the reason to compare them if they belong to the same breed?

Paper must be rewrite and should be made more clear. The tables must be more simple, the information not doubled. The daya are valuable but must be given in good form.

Reviewer 2 Report

Dear Authors,

Firstly, I would like to thank you for taking my suggestions into consideration. According to your statement, the title had been changed, but I did not see the change.

Yours sincerely

Round 3

Reviewer 1 Report

The obtained results are important aand valuable, but this paper is not possible to read. I would like to propose preparation of 2-3 separate papers about different topics. English is very difficult because and propoer words are not used. 

Some points below, but there could be many other comments and questions. 

Line 94 – it should be frequencies of genotypes and alleles

In Table 2 number of samples is smaller than number of genotypes ??? If there is 52 samples  and 1 sample has unique genotype the frequency will be 1/52 = 0.019 Values lower than this mean that genotype was of part of animal? Alleles are written differently – with or without subscript

In discussion below table 2 there is information about connections between blood group. It could be valuable to present relations of genotype/allele frequency to have possibility to discuss the epistasis.

Line 257 – what mean retired eves?

Line 272 – there is no information about slaughtered lambs in table

Tables 4 and 5 – why analyze the same locus for the same breed separately?

Table 6 – for what comparison the Chi square test is used?

Figure 1A and B – what is the relation between sample size and heterozygosity? How the t-test was done if for every breed there was one value?